# Position: Scale is a False Promise for Endangered Languages

Ivory Yang [1]   Soroush Vosoughi [1]

## Abstract

As endangered languages disappear, Machine Learning (ML) increasingly frames their revitalization as a problem of scale, emphasizing more data, larger models, and broader coverage. We posit that scale is not the limiting constraint in endangered language revitalization, and that progress lies in methodological and evaluative reorientation. Evidence from Language Identification (LID), Optical Character Recognition (OCR), and synthetic data generation shows that benchmark-driven scaling produces brittle or culturally misaligned outcomes, as evaluation and modeling lack epistemic fit. Advancement in this domain lies in rethinking methodology, by grounding evaluation in cultural fidelity, community trust, and situated use rather than abstract accuracy. The revitalization of endangered languages is not about the universality of success, but the specificity of care afforded to each language and community.

## 1. Introduction

Languages are not merely objects for documentation and evaluation, but living inheritances shaped by ritual, transmission, and collective memory (Fishman, 2001; Tkemaladze et al., 2025; Yang et al., 2025c). In recent years, the Machine Learning (ML) community has increasingly turned its attention toward so-called endangered languages (Mirza & Sundaram, 2016; San et al., 2021; Pinhanez et al., 2024), often framing them as sites of technical opportunity where performance can be stretched and generalization tested (Levow, 2019). Within this framing, endangered languages are commonly approached as optimization problems, addressed through transfer learning (Xia et al., 2021), fine-tuning strategies (Cruz & Cheng, 2019), or synthetic data

generation under minimal supervision (Hämäläinen, 2021).

These approaches rest on an implicit assumption, that *scale* is the primary constraint in endangered language revitalization. In this context, scale refers not simply to quantitative methodology, but a broader optimization paradigm of expanding data volume, benchmark coverage, synthetic generation, and model generality. While meaningful scaling, such as collecting high-quality and domain-relevant data, can improve performance (Assael et al., 2025), current approaches often rely on generalized increases in scale as a default solution. In practice, revitalization is often approached through a standardized pipeline: assembling web-scraped or archival data, expanding it through synthetic augmentation, training multilingual models, and evaluating success via downstream task performance (nll, 2024). When performance falls short, shortcomings are typically attributed to insufficient data coverage or model capacity, reinforcing the belief that further gains will follow from expanding scale (Mohanty et al., 2024). Yet evidence from Language Identification (LID), Optical Character Recognition (OCR), and synthetic data generation suggests that performance plateaus quickly, and that benchmark-driven scaling produces brittle or culturally misaligned outcomes (Caswell et al., 2020; Hill, 2023; Bender & Friedman, 2018; Yang et al., 2025d;e). These failures are not simply the result of limited data, but of evaluation and modeling practices that privilege abstract accuracy over trust, transmission, and cultural resonance (Cope & Penfield, 2011; Hale, 1992; Bird, 2024).

**We posit that scale is not the limiting constraint in endangered language revitalization, and that progress lies in methodological and evaluative reorientation.** In particular, this requires forms of epistemic integration that align ML objectives with community knowledge, sociolinguistic grounding, and situated use (Bird, 2020; Avineri & Kroskrity, 2014; Roche, 2019). More broadly, recent work has shown that ML systems often behave differently under low-resource (Yong et al.) and culturally distinct language conditions (Huang et al.), suggesting that these contexts may serve as stress tests for assumptions about generalization, transfer, and evaluation. From this perspective, endangered language revitalization is not merely a niche application domain, but a lens through which to examine broader methodological limitations in machine learning.

---

[1]Department of Computer Science, Dartmouth College, Hanover NH, United States. Correspondence to: Ivory Yang <ivory.yang.gr@dartmouth.edu>, Soroush Vosoughi <soroush.vosoughi@dartmouth.edu>.

*Proceedings of the 43$^{rd}$ International Conference on Machine Learning*, Seoul, South Korea. PMLR 306, 2026. Copyright 2026 by the author(s).

## 2. The Benchmark Fallacy

*Before offering this critique, we acknowledge the substantial and often underappreciated work already being done in endangered language research. Our perspective is shaped by sustained collaboration with language communities, working alongside native speakers and educators. Any effort that increases visibility or support for endangered languages is valuable. Our aim is not to dismiss such work, but to examine how prevailing methodological assumptions shape what ML research in this domain ultimately enables.*

### 2.1. Framing Languages as Metrics

The expansion of multilingual machine learning has brought renewed attention to low-resource languages (Dabre et al., 2020), often positioning them as stress tests for generalization (Levow, 2019). Benchmarks such as XTREME (Hu et al., 2020), Tatoeba (Anastasopoulos & Neubig, 2019), and FLORES-200 (nll, 2024) include endangered or underrepresented languages primarily as edge cases, useful for measuring performance degradation rather than as languages with distinct social, historical, and epistemic contexts (Liu et al., 2022). Within this framing, languages are reduced to token sequences to be translated or classified, and success is defined by metrics such as BLEU (Papineni et al., 2002) or perplexity rather than cultural relevance or real-world use (Gupta & Sharma, 2024).

We refer to this assumption as the *benchmark fallacy*: the belief that every endangered language can be meaningfully evaluated using the same tasks and metrics as high-resource languages. This paradigm privileges reproducibility and comparability while sidelining community priorities, sociolinguistic variation, and epistemic integrity (Anastasopoulos et al., 2020). Languages become testbeds for transfer learning and prompting strategies (Xia et al., 2021; Cruz & Cheng, 2019), rather than living systems embedded in intergenerational practice.

### 2.2. What Benchmarks Leave Out

Benchmark tasks reward what is easily quantifiable. As Bender et al. (2021) note in their critique of stochastic parrots, ML often treats phenomena that resist measurement as peripheral. This logic is particularly ill-suited to endangered languages, which frequently rely on oral transmission (Karyolemou, 2022), context-sensitive variation (DiCanio et al., 2013), and social norms of use that resist standardization (Awal, 2024).

Many Indigenous languages include seasonal vocabularies (Lavoie et al., 2014), ritual registers (Buckley, 1984), and taboo avoidance strategies (Kayangula, 2016) that vary across context, speaker, and social role (Hale, 1992). These features are central to language vitality, yet are largely absent from standard datasets (Mohanty et al., 2023). Available parallel corpora are often dominated by religious texts (McCarthy et al., 2020), which, while invaluable, reflect a narrow range of genres and communicative settings (Shahid et al., 2025). Models trained on such data may achieve high BLEU scores while producing outputs that lack cultural resonance or contextual appropriateness (Song et al., 2025; Leiter et al., 2024).

Table 1 summarizes this divergence. Benchmark-oriented approaches prioritize abstraction and scale, whereas epistemically grounded methods emphasize context, collaboration, and community-defined success.

### 2.3. Flattening Through Terminology

Terminology such as "low-resource," "underrepresented," or "minority language" further contributes to epistemic flattening by collapsing distinct sociopolitical realities into a single category of scarcity. For example, Ainu, a critically endangered language spoken in Northern Japan, which has an active revitalization movement but limited digital presence (Yotsumoto, 2020; Miyagawa, 2023), is labeled low-resource in much the same way as Maltese (Micallef et al., 2022). Yet Maltese is a national language with official EU status, institutional support, and substantial online content (Sciriha, 2002; 2024).

Such abstraction legitimizes the reuse of standard techniques, including transfer learning, synthetic data generation, and token realignment, without requiring researchers to ask what a language is for or whom it serves (Hämäläinen & Alnajjar, 2019; Zhang & Xu, 2022). In benchmarks such as FLORES-200, languages with vastly different histories and stakes are evaluated using identical metrics, obscuring more than it reveals. As Bird argues, this universalizing impulse risks reproducing extractive practices under the guise of technical neutrality (Bird, 2020).

### 2.4. The Myth of Emergence

The benchmark fallacy is further reinforced by Large Language Models (LLMs), which promise cross-lingual generalization through scale alone (Lai et al., 2024; Akavarapu et al., 2025). In this paradigm, endangered languages are expected to emerge from multilingual pretraining without fine-tuning or contextual grounding. When performance falls short, failure is attributed to insufficient data rather than to model design or evaluation choices (Alvarez et al., 2025). The assumption that all languages should be capturable by the same architecture and objective function often remains unexamined (Wein & Schneider, 2022).

*Table 1.* Comparison of benchmark-driven ML and epistemically integrated approaches to endangered language research.

| Research Area | Benchmark ML | Epistemic Integration |
|---|---|---|
| **Evaluation** | BLEU, perplexity, F1 | Community validation |
| **Data** | Web scraping, synthetic augmentation | Oral histories, speaker consultation |
| **Outputs** | Leaderboards, shared tasks, code | Teaching tools, community partnerships |
| **Assumptions** | Transfer learning, generalizability | Context-specific norms, sociolinguistic variation |
| **Collaboration** | Closed research community | Participatory design, shared authorship |

### 2.5. Interdisciplinarity Discouraged

Finally, benchmark-oriented research implicitly discourages interdisciplinary collaboration. Reproducibility (Belz et al., 2021), automatic evaluation (Resnik & Lin, 2010), and leaderboard comparability (Colombo et al., 2022) privilege forms of knowledge that are easily standardized. Local expertise, community judgment, and culturally grounded evaluation are treated as noise rather than signal. Unfortunately, many speakers and community experts of these languages are not members of the scientific community, and are therefore systematically excluded from research decisions about their own languages under standardized evaluation regimes.

This fallacy is not only epistemological, but ethical (Ishkhanyan, 2025). By optimizing surface-level token distributions, researchers may overclaim capabilities or deploy systems that appear fluent yet remain culturally hollow. In doing so, benchmark-driven approaches risk reinforcing the very dynamics of reduction and abstraction that have contributed to language endangerment in the first place (Blasi et al., 2022).

## 3. Evidence Against Scale

The limitations of benchmark-driven scaling are not theoretical. In this section, we present empirical cases in which increasing data volume or model capacity fails to produce meaningful gains for endangered languages. Across tasks, these examples show that performance often plateaus quickly, and that further scaling amplifies misalignment between machine learning objectives and the linguistic and cultural realities of language use.

### Case 1: Synthetic Data and Diminishing Returns

To compensate for data scarcity, recent work has explored the use of LLMs to generate synthetic data for endangered and low-resource languages (Patwa et al., 2024). However, evidence suggests that scaling synthetic data does not reliably translate into improved performance. A study on Named Entity Recognition (NER) across 11 low-resource languages from diverse families found substantial variability in gains from synthetic augmentation, with several languages showing negligible improvement (Kamath & Vajjala,

2025). Notably, small amounts of carefully curated, manually annotated data consistently outperformed large volumes of synthetic data, underscoring the limits of quantity-driven approaches.

Similar findings emerge in Neural Machine Translation (NMT). In Guarani-Spanish translation, grammar-based data augmentation improved performance relative to prior baselines, but only when linguistic plausibility and cultural relevance were explicitly enforced (Bartelds et al., 2023). Without such constraints, synthetic data risked introducing distortions. More broadly, recursive training on synthetic outputs has been shown to induce model collapse, degrading performance as errors compound across generations (Shumailov et al., 2024). Together, these results suggest that beyond a small threshold, additional synthetic data offers diminishing returns and may actively undermine model quality. Synthetic data expansion is not a crutch for endangered languages to rely on to outperform benchmarks. It should be used sparingly, and in some cases, not at all.

### Case 2: Language Identification at Scale

Language Identification (LID) systems are often presented as evidence that scale enables broad linguistic coverage. Widely used tools such as the Google LangID API (Google Cloud, 2025), fastText LID-176 (Joulin et al., 2017), CLD3 (Google Inc.), LangID.py (Lui & Baldwin, 2012), and the WiLI-2018 benchmark (Thoma, 2018) report support for up to hundreds of languages. Yet, as shown in Table 2, coverage of endangered and Indigenous languages remains sparse.

More importantly, increased coverage does not guarantee reliability. Large-scale systems routinely misidentify underrepresented languages, particularly those that are typologically distinct or sparsely represented in training data (Alvarez et al., 2025; Yang et al., 2025b). For example, phonemized Navajo was misclassified by Google's LID system as unrelated languages such as Icelandic or Lingala (Yang et al., 2025d). This phenomenon reflects a twofold structural problem; although Navajo is not supported as a language option in Google Translate, the LID system does not flag the input as unsupported, but misclassifies it as typologically unrelated languages. On the other hand, the

*Table 2.* Reported language coverage of commonly used language identification systems.

| System | Reported Languages | Endangered Language Coverage |
|---|---|---|
| Google LangID API | ∼100 | Limited coverage, primarily major world languages. Lacks support for most Indigenous and endangered languages. |
| fastText (LID-176) | 176 | Partial coverage, including Haitian Creole, Zulu, Welsh, and Maori. Minimal representation from the Americas, Australia, and Arctic regions. |
| CLD3 | 107 | Very limited coverage of endangered languages; supports Zulu but omits most Indigenous languages. |
| LangID.py | 97 | Limited coverage, including Welsh and Maori, with major gaps across the Americas and Australia. |
| WiLI-2018 Benchmark | 235 | Variable coverage, with inconsistent representation of endangered and Indigenous languages. |

exclusion of Navajo itself from Google Translate is concerning, given that Navajo is the most widely spoken Native American language in the United States, with over 170,000 speakers as recorded in the 2019 census. Caswell et al. (2020) report similar brittleness in large web-scale corpora construction, where LID failures persisted despite extensive data collection and preprocessing, reflecting a mismatch between model assumptions and linguistic reality.

### Case 3: OCR Without Contextual Grounding

Digitization is often presented as a neutral or universally beneficial step in endangered language preservation. In practice, however, digitization efforts for endangered languages frequently rely on Optical Character Recognition (OCR) systems designed for high-resource scripts, typographic conventions, and institutional archives. General-purpose OCR models continue to struggle with nonstandard orthographies, rare or defunct fonts, extended Latin characters, and multilingual layouts that are common in endangered language materials (Rijhwani et al., 2020; Agarwal & Anastasopoulos, 2024), even as model size and training data increase.

Crucially, many endangered language texts do not resemble the clean, monolingual documents assumed by benchmark OCR settings. Instead, they originate from historical archives or community-led documentation efforts and often include handwritten marginalia, idiosyncratic typefaces, interlinear glosses, and layout conventions shaped by pedagogical or revitalization needs rather than print efficiency (Ng et al., 2009; Schillo & Turin, 2022; Gupta et al., 2006). Off-the-shelf OCR systems frequently flatten or disrupt this structure, breaking alignment between parallel annotations and erasing contextual cues that are essential for interpretation and reuse (Fleischhacker et al., 2025). Correcting these errors requires extensive post-processing that is not only labor-intensive, but also demands cultural and linguistic expertise rarely accounted for in system evaluation.

Taken together, these failures illustrate a recurring pattern:

without contextual and cultural grounding, scaling OCR models does not address the core limitations faced by endangered language (Perera et al., 2025; Yang et al., 2025a). Instead, it risks reproducing the same abstraction-driven assumptions that marginalize the very practices and knowledge systems these digitization efforts are intended to support. Quality over quantity favors specialized, script-specific OCR over further scaling of generalized models.

### Summary

Across synthetic data generation, LID, and OCR, these cases demonstrate a consistent pattern: initial gains are possible with modest data, but further scaling yields diminishing returns and increasingly brittle behavior. Scale is a false promise, and these findings reinforce the need to move beyond scale as the dominant strategy for progress.

## 4. Toward Epistemic Integration

If scale is not the limiting constraint in endangered language revitalization, then progress requires rethinking how machine learning systems are designed and evaluated. We propose *epistemic integration* as a response to the limitations of benchmark-driven approaches: the deliberate incorporation of community knowledge, sociolinguistic grounding, and cultural context into machine learning practice. This is not a rejection of rigor or reproducibility, but an expansion of what rigor entails when working with socially embedded languages. Table 3 summarizes practical strategies informed by fieldwork and prior collaborations.

### 4.1. Matching ML Paradigms to Cultural Contexts

Endangered language communities differ widely in their linguistic practices, pedagogical traditions, and revitalization goals. A single modeling paradigm cannot serve all contexts equally well. Approaches that emphasize universal multilinguality or language-agnostic benchmarks (Zhou et al., 2022; Feng et al., 2022) often flatten these differences, prioritizing

*Table 3.* Culturally grounded machine learning strategies inspired by past fieldwork.

| Region | Languages | Key Traits | Recommended Approach |
| --- | --- | --- | --- |
| Americas | Navajo, Quechua | Oral tradition, language sovereignty | Co-design with communities; prioritize trust, continuity, and pedagogical use |
| Africa | Kinyarwanda | Local variation, grassroots networks | Fine-tune with speaker input; invest in shared and locally maintained tools |
| Asia | Old Chinese, Konkani | Ceremonial use, metaphor-rich expression | Culturally informed OCR; respect orthographic and semantic idiosyncrasies |
| Europe | South Jutlandic, Neapolitan | Dialect pride, stylistic variation | Style-aware generation; support expressive and dialectal nuance |

generality over cultural compatibility.

We argue for culturally contextual ML design, where modeling choices reflect how language is used within a community (Wein, 2025). This includes whether a language is primarily oral or textual, whether ceremonial registers are preserved, and how data sovereignty is negotiated. These considerations should shape architectures and evaluation protocols rather than being treated as downstream concerns.

## 4.2. Design with Purpose

Work on endangered languages should begin by asking what a system is intended to support, and for whom (Liu et al., 2022). Task definitions imported from high-resource settings often misalign with local needs. For some communities, the priority may be ceremonial preservation or documentation (Siregar, 2022); for others, intergenerational learning or dialectal expression (Han, 2023).

Designing with purpose requires early involvement of speakers, linguists, and cultural practitioners. When goals are defined collaboratively, ML systems are more likely to support meaningful use rather than abstract performance. A practical step would be for large technology developers to emulate initiatives such as Meta's Language Technology Partnership Program (LTTP) (Omnilingual et al., 2025), creating structured programs that combine scale with community partnership rather than treating inclusion as an afterthought.

## 4.3. Evaluate for Relevance

Standard metrics such as BLEU (Papineni et al., 2002) or ROUGE (Lin, 2004) are insufficient indicators of success in endangered language contexts. These measures reward surface regularities while penalizing variation, creativity, and stylistic nuance. We advocate for evaluation practices that account for relevance rather than abstraction, including community validation, cultural fidelity, and alignment with revitalization goals (Liu et al., 2022).

Plural evaluation requires aligning evaluation methodology with the linguistic structure of individual languages, rather than assuming uniform units of comparison. For exam-

ple, several Native Alaskan languages such as Yup'ik are polysynthetic (Park et al., 2021), meaning that what corresponds to an entire sentence in English may be expressed as a single word. Under standard token-level metrics such as BLEU or ROUGE, valid translations may therefore be systematically penalized despite preserving the underlying grammatical and semantic content. In such settings, evaluation at the morpheme or character level, or assessment of whether core semantic and grammatical structures are correctly realized, may provide a more appropriate measure of quality. This illustrates that evaluation for endangered languages cannot always rely on monolithic benchmarking frameworks, but must adapt to the linguistic and cultural properties of each language community.

Plural evaluation also extends beyond metric design alone. Determining whether a system meaningfully supports language use may require direct participation from speakers and community stakeholders themselves, particularly in cases where appropriateness, usability, or cultural resonance cannot be fully captured through automatic evaluation (Dent et al., 2024; Sälevä & Lignos, 2026). As an example, the Mayan community has been a forerunner in engaging with the ML community, having its own delegation at the 2025 UNESCO Language Technologies for All (LT4All) meeting (Tomas, 2025). Similar representation could be replicated with additional language communities, enabling speakers themselves to assess system outputs for appropriateness, usability, and cultural alignment using community-defined criteria. While the 2025 LT4All marked a long-awaited renewal of international coordination following the 2019 iteration, its scale also highlighted the substantial logistical, institutional, and political overhead required to convene such efforts, making frequent repetition difficult. A complementary and more sustainable approach lies in regionally organized or community-led forums, analogous to how large research communities often distribute participation across regional venues operating under a shared framework rather than concentrating engagement within a single global forum. Although these setups require time to establish and resist full standardization, they provide a more faithful measure of whether a system meaningfully supports language use.

### 4.4. Collaborate in Authorship

If scale is no longer the primary lever for progress, then collaboration becomes central. Research on endangered languages must be conducted with communities rather than on them. This includes shared authorship, participatory design, transparent data practices, and respect for data sovereignty.

In practice, this means expanding who is recognized as a legitimate research contributor and how collaboration begins. Affinity groups such as Indigenous in ML (NeurIPS) have created important spaces for affiliated researchers already working within the field; however, many language communities remain structurally excluded long before this stage. Collaboration can begin earlier by engaging students and community members who are already interested in research, including high school and undergraduate learners seeking hands-on experience. Community-embedded mentorship and co-authorship models not only lower barriers to entry, but also foster "homegrown" researchers who are trusted locally and invested in long-term stewardship of their language (Gewin, 2021).

*We have increasingly been contacted by students from endangered language communities seeking guidance on research related to their own languages. They bring what is often hardest to obtain, including language knowledge and community consent, but lack access to traditional academic research pathways. While we have chosen to volunteer when possible and have successfully mentored several publications, this gap does not scale in the long run. Addressing it would be better served by a central, impartial authority rather than individual researchers, such as by extending structural models already used for mentored early-stage research participation (e.g. Student Research Workshop (SRW) pre-mentorship system).*

### 4.5. Avoid Savior Narratives

Finally, ML should not be framed as a approach that can *save* endangered languages. Revitalization is a community-led, generational process shaped by social relations and cultural stewardship (Bird, 2020). ML can play a supportive role only when it defers to community priorities and acknowledges its limits (Meissner, 2018).

Humility is therefore not ancillary, but methodological. Treating scale as a solution risks reproducing extractive dynamics; treating care as a constraint opens space for more responsible engagement. Accordingly, just as how most ML venues now include an explicit ethics statement or check for paper submissions (Ashurst et al., 2022), work situated in endangered language and Indigenous language contexts should be accompanied by a required framing review or checklist that explicitly assesses tone, narrative positioning, and claims of impact, with the goal of identifying and preventing savior-oriented or extractive representations.

## 5. Alternative Views

We recognize that the position advanced here challenges prevailing assumptions in ML. Below, we address common counterarguments and explain why they are insufficient in the context of endangered languages.

### 5.1. "Benchmarks Necessary for Scientific Progress"

A recurring concern is that abandoning benchmarks risks undermining scientific rigor. This concern is not unfounded. Standardized assessments serve important functions: they enable comparability across systems, establish minimum competency thresholds, and provide decision-making signals in high-stakes contexts (Clark & Etzioni, 2016). Much like standardized tests in education, benchmarks can function as coarse but necessary indicators when some form of scoring is required (Vo et al., 2025). The question is not whether benchmarks are ever useful, but whether they are sufficient as the primary or exclusive measure of success.

Benchmark-oriented scaling has undeniably produced substantial advances in domains where abstraction and generalization are appropriate objectives (Chen et al., 2024). For example, large-scale graph representation learning has demonstrated strong performance gains through unified sparsity and multi-granular semantic modeling (Zhang et al.; Wen et al., 2024), while hierarchical representation learning has improved embodied decision-making in agentic environments (Zhang et al., 2025). These successes illustrate that scaling-oriented paradigms are not inherently flawed. Rather, our argument is that endangered language revitalization presents a fundamentally different setting, where sociolinguistic grounding, cultural stewardship, and contextual relevance cannot be reduced to benchmark optimization alone.

Benchmarks have accelerated progress by enabling reproducibility and comparison. However, their dominance can narrow what counts as valid contribution. In endangered language research, benchmark-friendly task definitions often distort revitalization goals and exclude context-sensitive work (Awal, 2024). Scientific rigor does not always require uniform metrics. Fields such as Human-Computer Interaction (HCI) and education employ pluralistic evaluation frameworks that reflect user goals and contextual variation (Whitefield et al., 1991). ML applied to language should adopt a similarly flexible view.

### 5.2. "Community Collaboration Slows Down Research"

Concerns about the pace of community collaboration are not without merit. Collaborative processes introduce additional coordination costs, extend project timelines, and may

limit the scope of technically feasible experimentation (Fernando et al., 2010). In fast-moving research environments, such constraints can appear at odds with prevailing incentive structures (Shethiya, 2023). Moreover, not all communities view computational intervention as desirable: Indigenous and minoritized language communities have had (fairly valid) reservations towards participation in ML-driven language projects (Jones et al., 2025), citing concerns over data sovereignty, misrepresentation, or historical extraction. These responses underscore that access and consent cannot be assumed, nor can speed be treated as a neutral objective.

Collaboration requires time, trust, and negotiation. Yet this slowness is not a liability but a safeguard. Faster pipelines that exclude stakeholders may yield publishable results, but they risk producing tools that are unused, mistrusted, or harmful in practice. Collaborative work when implemented correctly (Bird, 2024) is often more durable and less prone to deployment failure, precisely because it accounts for context from the outset (Cowell et al., 2023).

### 5.3. "Machine Learning is Apolitical"

The appeal of framing ML as apolitical is understandable. Abstraction enables generalization, and formalism allows systems to be evaluated independently of the social contexts in which they are deployed (Zimmerman et al., 2025). In many domains, this separation has been productive, allowing technical advances to be shared across applications without requiring domain-specific adjudication. However, this assumption breaks down when the object of modeling is itself socially situated, historically marginalized, or subject to ongoing power asymmetries (Song, 2022).

Claims of neutrality obscure the fact that modeling choices shape visibility, authority, and access (Manić, 2026). Decisions about which languages to support, which metrics to optimize, and whose judgments count are inherently political (Voort, 2024; Ugochukwu, 2025). Endangered languages have been marginalized through historical processes of suppression and neglect (Roche et al., 2023). Modeling them without acknowledging this context risks perpetuating epistemic erasure rather than mitigating it.

## 6. Recommendations

If scale is not the primary constraint in endangered language revitalization, then progress depends on how ML research is designed, evaluated, and supported. We offer the following recommendations for key stakeholders in the ML ecosystem.

### 6.1. For Machine Learning Researchers

- **Reframe task design**: Begin with community needs and linguistic context rather than predefined NLP tasks. Consider goals such as intergenerational storytelling,

pedagogical support, or ceremonial documentation.

- **Adopt plural evaluation**: Complement quantitative metrics with community-informed assessments of cultural fidelity, trust, and relevance. It is imperative to include actual speakers and community members as part of the evaluation process.

- **Practice epistemic humility**: Clearly articulate what models can and cannot capture, and document potential risks, limitations, and misuse.

- **Document data and ethics rigorously**: Provide detailed data statements, consent practices, and ethical review processes, particularly when working with human subjects or culturally sensitive materials.

### 6.2. For Conferences and Reviewers

- **Broaden definitions of rigor**: Value submissions that employ non-standard evaluation, community validation, or interdisciplinary methods.

- **Recognize collaborative labor**: Encourage and reward shared authorship with linguists, local scholars, and community members. Potentially, have a specialized track or venue to receive submissions for such collaborations.

- **Review for epistemic fit**: Assess whether the framing, methods, and evaluation align with revitalization goals rather than relying solely on benchmark performance.

### 6.3. For Funders and Institutions

- **Support infrastructure beyond models**: Invest in long-term partnerships, mentorship programs, documentation workflows, educational tools, and annotation pipelines co-developed with communities.

- **Reward process alongside outcomes**: Recognize the value of collaboration, trust-building, and ethical engagement, even when progress is slower or less easily quantified.

- **Protect language sovereignty**: Establish policies that ensure funded work respects community autonomy and long-term priorities.

Together, these recommendations reflect a shift from scale to stewardship, from performance to purpose, and from extraction to reciprocity.

## 7. Limits, Tradeoffs, and the Value of Partial Success

A common critique of community-centered and epistemically grounded approaches is that they may be overly

idealistic or insufficiently scalable. Collaborative work is slower, context-sensitive evaluation resists standardization, and outcomes may be uneven or incomplete. From a purely efficiency-oriented perspective, such constraints can appear to limit impact.

Yet many domains recognize that partial success does not imply diminished value. In conservation biology, efforts to preserve endangered species continue even when extinction cannot be universally prevented, because maintaining biodiversity where possible is preferable to abandonment (Lindenmayer & Hunter, 2010). In cultural heritage preservation, resources are devoted to restoring manuscripts, artworks, or architectural sites despite the inevitability of loss and decay (De Beer & Boogaard, 2017). Similarly, in public health and disaster response, interventions are justified even when they cannot reach all populations or eliminate all harm (Parsons & Neath, 2017).

Endangered language revitalization operates under comparable conditions. Not all languages can be saved (Sperlich & Uriarte, 2015), and not all revitalization efforts will succeed (Costa, 2024). However, this reality does not invalidate the work. On the contrary, it underscores the importance of directing care, resources, and methodological attention where they can meaningfully support transmission, use, and community goals. The alternative is not neutrality, but neglect. From this perspective, ML research that prioritizes epistemic fit over scale is not an exercise in idealism, but a pragmatic response to structural limits. When universal solutions are unavailable, value lies in specificity. Supporting some languages well is preferable to supporting many languages superficially. Progress, in this domain, is measured not by coverage alone, but by whether interventions matter in practice (O'Grady, 2018; Flavelle & Lachler, 2023).

*As a position paper, this work does not claim omniscience, nor does it presume the capacity to unilaterally enact the structural changes it argues for. Nonetheless, this conversation is both timely and necessary. Given the current trajectory of the field, raising these questions from a position of sustained engagement is not an assertion of final answers, but an acknowledgment of responsibility. Silence, at this moment, would be the more consequential choice.*

## 8. Conclusion

Over the years, significant breakthroughs have been achieved in language revitalization, yet the dominant ML approach toward endangered languages remains shaped by assumptions that do not hold at the margins. By treating endangered languages as benchmark problems, the field has overemphasized scale while underinvesting in methodological alignment, evaluative relevance, and community engagement. **We posit that scale is not the limiting constraint in endangered language revitalization, and that progress lies in methodological and evaluative reorientation.**

Through empirical cases and practical guidance, we have demonstrated that benchmark-driven approaches and scaling often produce diminishing returns, while epistemically grounded methods offer more meaningful pathways. The revitalization of endangered languages is defined not by universally quantifiable success, but by the specificity of care afforded to each language and community. Attending to these margins does more than support revitalization efforts, as it exposes the limits of prevailing assumptions, and challenges the ML community to refine what it means by progress and understanding. *Non multa, sed multum* is the way forward.

## Acknowledgements

This work was partially supported by a CompX Faculty Grant from the Neukom Institute for Computational Science at Dartmouth College. The first author would also like to thank Dr. Joseph Mariani for his mentorship and support, particularly for the opportunity to present at the UNESCO Language Technologies for All (LT4All) conference, which broadened her perspective on language revitalization efforts and the global language technology community.

## Impact Statement

This paper presents work whose goal is to advance the field of Machine Learning. There are many potential societal consequences of our work, none which we feel must be specifically highlighted here. Ultimately, it is our hope that this work contributes meaningfully to ongoing efforts in endangered language revitalization.

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
