# OpenReview forum: "Position: Scale is a False Promise for Endangered Languages"
_ICML.cc/2026/Position_Paper_Track — ICML 2026 Position Paper Track regular_

### Official Review · Reviewer_qayy · 2026-03-10

**Significance:** 4
**Argument Clarity:** 4
**Rating:** 4
**Confidence:** 3

**Questions:**

1. Do the authors think that the current benchmarks for low-resource languages are not sufficient in capturing the nuances required within those languages, or that benchmarks inherently cannot capture the nuances required in dealing with low-resource languages? For example, would a carefully-enough designed benchmark be able to quantify whether the LLM can respect the context or characteristics to be learned sufficiently well? I am wondering whether the characterisation of benchmarks in this context could have been expanded to include something more idealistic, and more in line with objectives the paper aims to achieve.

2. It can be argued that construction of benchmarks to reflect what the community wants to gear towards is what allows for more interest within the sub-area. A typical researcher may not be able to easily seek out language experts to collaborate with, and using published benchmarks (possibly constructed _by_ the language experts) may be an easier way to enter said community to verify different ideas, even if not to publish or deployed in practice. Of course, this wouldn't discount the fact that results from benchmarks should be treated with caution. I also get that this point is responded to in Sec 5.2 by the authors, however I am also doubtful about to what degree gatekeeping these areas of research may have the effect of over-discouraging research in these areas.

**Alternative Views Section:**

Yes

**Compliance With Llm Reviewing Policy A Conservative:**

Affirmed.

**Discussion Potential:**

3

**Final Justification:**

I maintain my positive score, since any of the weaknesses I feel this paper has is mainly to do with the writing or additional viewpoints I think could be considered (which have been addressed by the authors in their responses). The key message from the paper is already quite good even prior to the rebuttals from the authors, and is worth a venue for sharing.

**Paper Summary:**

The paper argues that in low-resource languages, the typical assumption is that there is not enough data, which leads to how the methods dealing with these problems are proposed, or diagnosed when performance suffers. The paper then says that this is not the case, and that in practice, much research is too benchmark-oriented, such that it claims gains that do not necessarily translate to practical performances and that would be accepted by the community familiar with such language. These claims are backed by several case studies of problems which are unable to give good results when used on low-resource languages in practice.

**Position:**

Yes

**Position In Title:**

Yes

**Related Work:**

3

**Strengths And Weaknesses:**

Strengths:

- I find this to be a relevant viewpoint and discussion-worthy in the context of LLMs, especially when there is much discussion about how LLM research is currently too rapid and that the works published may not be in line with what is really needed. The paper seems to present one sub-point/case study within this larger debate.

- The paper is quite well-structured, and discusses the issues clearly with related evidence (from case studies of different problems that don't work well, to current scenarios where people are taking alternate approaches to handling low-resource languages).

- I feel the Alternate Views are quite well written, as it tackles the counter-arguments of the works directly and pre-emptively argues against them well enough.

Weaknesses:

- I am wondering if the characterisation of benchmarking could be elaborated better in the paper. In modern LLM research we do employ a narrower definition of benchmarks (in that benchmarks do typically measure one specific aspect of language understanding), but it might be unclear whether this is due to benchmarks not being well-designed or that they inherently can only be used for specific classes of language problems.

- The term "scale" in the paper could also be better elaborated. I understand that it seems to be referring to the increasing amount of data or model capacity, and that much of the problems in the paper deals with the inability for us to scale up our methods in the low-resources language space. But increasing the amount of irrelevant data may not be considered true scaling, and rather trying to increase the data in the correct domain is a more "real" sense of scaling. I guess the point may be more that trying to scale up general-purpose LLMs to work across all languages is unhelpful for these low-resource languages -- which could possibly be made more clear as the premise behind the works and the evidence (and to what scaling is being considered).

- I think the claim that "scale is not the limiting constraint in endangered language revitalization" could be better backed, or rephrased to reflect better what the authors may be trying to convey.

  - The paper shows example of cases where under limited "scale" (i.e., less training data to learn from), the existing methods in learning from less resources don't show improvement. However, it doesn't directly show that these issues wouldn't be alleviated if we did have more data. Another way I see it is -- if the low-resource languages did have data on the scale that a language like French may have, would we still have problems that things don't work well (i.e., any inherent difficulties with them now)?

  - I do agree methods developed by researchers may not align with what is needed according to the experts in said low-resource languages, and that these sort of changes to look at how languages should be modelled based on experts are in fact important (and may in fact speed up progress in those sub-areas). The position of text in bold may not necessarily be reflecting this currently, so it likely is a matter of rewording that part of the position. It may be related to the previous point about how "scaling" is defined.

**Support:**

3

---

> ### Author Rebuttal · Authors · 2026-03-28
>
> We appreciate the reviewer’s insightful comments.
>
> Regarding benchmarking, we agree with the reviewer’s distinction. Our intention is not to suggest that benchmarking is inherently incapable of capturing relevant linguistic properties, but rather that many current benchmarks used in low-resource settings remain insufficiently aligned with the linguistic structure and community goals of endangered languages. For example, in polysynthetic languages such as Yup’ik, where a single word can encode what would correspond to an entire sentence in English, common metrics such as BLEU or ROUGE may fail to adequately reflect translation quality. We will clarify that our proposed notion of plural evaluation includes not only broader use of metrics, but also better-designed, community-informed benchmarks that are more responsive to linguistic and cultural context, rather than relying on a single monolithic evaluation framework. This was an excellent point, and we appreciate the reviewer for highlighting it.
>
> On the notion of “scale,” we also appreciate the reviewer’s clarification and will adopt a more precise definition in the revision. In particular, we distinguish between meaningful scaling, such as increasing high-quality, domain-relevant data, and superficial scaling, such as expanding datasets through synthetic generation or unrelated corpora. Our concern is that current approaches can sometimes treat the latter as evidence of progress, even when it produces limited gains in alignment, utility, or performance. We will therefore revise the paper to make clear that our argument is not against scaling in itself, but against relying on general-purpose increases in model or data scale as a default solution for endangered language settings.
>
> We agree that our statement that “scale is not the limiting constraint” is too strong in its current form. Our intended claim is narrower: in many endangered language contexts, scaling alone is not a sufficient solution, particularly where high-quality data is inherently scarce, finite, or costly to produce. In such settings, progress may depend more critically on alignment with linguistic structure, careful data collection practices, and appropriate evaluation. We will revise this statement to better reflect that nuance.
>
> Finally, we appreciate the reviewer’s concern regarding researcher participation. Our goal is not to gatekeep participation, but to encourage broader and more responsible engagement. In the same way, we do not argue that benchmarks are wholly unconstructive, but rather that they must be applied carefully and in ways that can coexist with meaningful community participation. We view recent trends, including increased participation from speakers within language communities, as a highly positive development made possible in part by the growing accessibility of ML tools. In this context, our aim is to advocate for approaches that make such participation more effective, respectful, and well aligned, rather than restricting entry into the field.
>
> Thank you again for the care and consideration you have demonstrated in your evaluation of our work.

---

> > ### Author Rebuttal · Reviewer_qayy · 2026-04-02
> >
> > I am quite okay with the responses from the authors, and additionally with the revision the author will consider for their paper. I will maintain my positive score of the paper.

---

### Official Review · Reviewer_E4SQ · 2026-03-12

**Significance:** 2
**Argument Clarity:** 3
**Rating:** 5
**Confidence:** 3

**Questions:**

Regarding suggestions "for conferences and reviewers", NLP community has already initiated efforts in this direction. Should ML community also consider adopting similar initiatives simultaneously?

I'd consider claiming something like data-driven or ML-driven "benchmarking and evaluation" is insufficient for endangered language, more directly. I believe ML is an open community for concrete guidance.

**Alternative Views Section:**

Yes

**Compliance With Llm Reviewing Policy A Conservative:**

Affirmed.

**Discussion Potential:**

2

**Final Justification:**

I'm convinced with the response and I believe it will be invaluable to ML community.

**Paper Summary:**

This paper argues that scale is not the limiting constraint in endangered language revitalization, and that progress lies in methodological and evaluative reorientation.

**Position:**

Yes

**Position In Title:**

Yes

**Related Work:**

3

**Strengths And Weaknesses:**

Strengths:

+ The claim is rational and supported with understanding in both ML and Endangered Language research.

+ The paper tries to connect the practice in ML with Epistemic Integration. This could be very inspiring, though the possible practical actions for MLers are still vague at the moment.

Weaknesses:

- The position is a long-existing situation, since the traditional statistical machine learning approach for low-resource languages or possibly even earlier.

- It would be beneficial to explicitly pinpoint the ML technology that might be helpful for the integration. Many ML researchers and engineers are eager to help, though short of a clear framework to execute or explore. For one example, it is good to know that we should "adopt plural evaluation", then what would be the technical suggestion, and how such evaluation connects to linguistic research?

- Lack of discussion on the impact if we do not solve the issue.

**Support:**

3

---

> ### Author Rebuttal · Authors · 2026-03-28
>
> We appreciate the reviewer’s established domain knowledge and thoughtful suggestions.
>
> First, we agree that this position has long existed in low-resource language settings. However, we argue that it is particularly important to revisit this problem at this moment. Recent advances in large-scale multilingual modeling, along with increased accessibility to ML tools such as open-source large language models (LLMs) and API credits, have significantly democratized low-resource language research. Whereas such work previously relied on ML researchers collaborating with language communities, we are now seeing more native speakers leading research on their own languages, including work on Cherokee [1] and Kinyarwanda [2]. This shift creates substantial new opportunities, but also introduces risks: as these tools become more widely available, there is a growing need for clearer guidance on how such research should be conducted responsibly and effectively.
>
> We acknowledge the reviewer’s request for more concrete guidance. One definition of our “plural evaluation” is to align evaluation metrics with the linguistic structure of each language. For example, several Native Alaskan languages such as Yup’ik are polysynthetic, meaning that entire sentences may be expressed as single words. Under standard token-level metrics such as BLEU, this would systematically penalize valid translations, as a single Yup’ik word may correspond to an entire English sentence. In such cases, evaluation at the morpheme or character level, or assessment of whether the underlying grammatical and semantic components are correctly realized, would be more appropriate. This example illustrates how evaluation must adapt to linguistic structure rather than assume uniform units of comparison across languages. We will include additional concrete examples in the Appendix. At the same time, we emphasize that a single universal technical approach is neither feasible nor desirable, as appropriate evaluation depends on language-specific linguistic properties.
>
> Finally, we agree that clarifying the consequences of inaction is important. Beyond the loss of linguistic and cultural knowledge, endangered language settings are not only sites of application but also sources of insight for improving ML systems more broadly, for example in understanding model behavior under low-resource or adversarial conditions such as jailbreaking [3]. Failing to engage with these settings risks not only the loss of languages, but also the loss of potential scientific insights. More broadly, as AI systems become increasingly embedded in society, neglecting low-resource languages risks deepening the divide between those who benefit from AI and those who do not.
>
> Regarding current initiatives, while the NLP community has made visible contributions, the ML community has also begun to engage through efforts such as “Indigenous in ML” [4] at NeurIPS. These efforts highlight growing awareness, but systematic solutions remain limited. We appreciate the reviewer’s suggestion to more explicitly state that standard benchmarking and evaluation paradigms are insufficient for endangered languages, and will incorporate this clarification in the revision.
>
> We hope our detailed clarifications help address the reviewer’s concerns and reflect our sincere effort to strengthen the paper in line with this feedback.
>
> [1] ChrEn: Cherokee-English Machine Translation for Endangered Language Revitalization, EMNLP 2020
>
> [2] KinyaBERT: a Morphology-aware Kinyarwanda Language Model, ACL 2020
>
> [3] Low-Resource Languages Jailbreak GPT-4, NeurIPS SoLaR 2023
>
> [4] Indigenous in AI/ML: NeurIPS 2025 Affinity Event. NeurIPS, 2025

---

> > ### Author Rebuttal · Reviewer_E4SQ · 2026-04-03
> >
> > The response resolved my concerns and I believe this work is invaluable to the ML community. I'll raise my score to accept!

---

### Official Review · Reviewer_mgMd · 2026-03-13

**Significance:** 3
**Argument Clarity:** 2
**Rating:** 5
**Confidence:** 3

**Questions:**

NA

**Alternative Views Section:**

Yes

**Compliance With Llm Reviewing Policy A Conservative:**

Affirmed.

**Discussion Potential:**

3

**Final Justification:**

Most weaknesses I saw in this paper have been satisfactorily addressed in the rebuttal. The position relates to a somewhat niche topic but that's a minor issue and I'm thus keeping my positive score.

**Paper Summary:**

The position advocated in this paper is that scaling (in the sense of larger datasets, larger models and broader coverage) is not the main constraint on efforts to revitalize endangered languages. Instead, the authors argue, focus should be more on engaging with and involving the community which speaks it, and paying more attention to the cultural context. Three cases are put forward to illustrate the "scaling" approach failing: use of synthetic data, language indentification, and OCR.

**Position:**

Yes

**Position In Title:**

Yes

**Related Work:**

3

**Strengths And Weaknesses:**

The paper states the position explicitly, and supports it with cases studies and logical argumentation. As such it is a good fit to the position paper format. Regarding relevance, language revitalization has substantial sociatal relevance, though it is not parricularly central to the ICML community. In this sense, it may be of narrow relevance.

The argument in the paper has a couple of weaknesses. Firstly, the concept of scale is overly broad. It seems that anything involving data-driven methodology and quantitative benchmarking is considered as "scale". I'm not sure this is the most useful framing.
Secondly the writing style is not very accesible, relying on a lot of jargon. The argument could be presented in a simpler, more straighforward way which would make it more understandable to a broad audience.

**Support:**

3

---

> ### Author Rebuttal · Authors · 2026-03-28
>
> We thank the reviewer for their thoughtful feedback.
>
> On the question of relevance, while language revitalization may appear niche, we emphasize that endangered languages expose broader limitations in current ML paradigms. Recent work has shown that models behave differently under low-resource [1] and culturally distinct language settings [2], suggesting that these contexts could be valuable for stress-testing assumptions about model generalization. Endangered language revitalization is therefore not an isolated application, but a lens through which to examine fundamental challenges in ML. We will make this broader relevance more explicit in the revised paper.
>
> Regarding scale, we clarify that we use the term to refer to several distinct but related practices: (i) uncritical data expansion through synthetic generation (Section 3: Case 1), (ii) extending benchmarks to cover languages that cannot be meaningfully supported in practice (Section 3: Case 2), and (iii) applying monolithic, quantitatively driven approaches across diverse languages without accounting for linguistic and cultural differences. We position this form of scaling as increasingly recognized to be problematic, and our goal is to argue for more context-sensitive alternatives. We agree that this framing could be made clearer, and will refine the definition in the revision.
>
> We also appreciate the comment on jargon. We agree that some terminology may be overly domain-specific and will revise the paper to improve accessibility, including modifying phrasing and validating readability with ML researchers outside this area.
> Thank you again for your constructive suggestions. We believe incorporating them will strengthen the paper, and we hope this clarification helps support a more favorable overall assessment from you.
>
> [1] Low-Resource Languages Jailbreak GPT-4, NeurIPS SoLaR 2023
>
> [2] Obscure but Effective: Classical Chinese Jailbreak Prompt Optimization Via Bioinspired Search, ICLR 2026

---

> > ### Author Rebuttal · Reviewer_mgMd · 2026-04-01
> >
> > The following concerns have mostly been resolved by the rebuttal, in the sense that the authors promise to address them, and that they can be reasonably addressed: clarification of the concept of scale, and improving readability by ICML audience.
> >
> > The point about relatively narrow relevance of language revitalization to the ICML audience: I don't think the authors can do all that much to address this (although it is good to connect this issue to broader problems in ML). At the same time I don't think this is a necessarily a shortcoming of the paper: some topics are simply less popular or less central to ICML audience than others, and that's fine.

---

### Official Review · Reviewer_yKkf · 2026-03-14

**Significance:** 3
**Argument Clarity:** 3
**Rating:** 4
**Confidence:** 2

**Questions:**

See above

**Alternative Views Section:**

Yes

**Compliance With Llm Reviewing Policy A Conservative:**

Affirmed.

**Discussion Potential:**

3

**Final Justification:**

The authors have addressed my concerns and I will maintain the positive score.

**Paper Summary:**

The paper posits that existing ML approaches incorrectly treat endangered languages as a problem of scale, aiming for larger models and more data. They then propose benchmark fallacy that drives brittle or culturally misaligned outcomes. Finally, the authors advocates for epistemically grounded methods.

**Position:**

Yes

**Position In Title:**

Yes

**Related Work:**

3

**Strengths And Weaknesses:**

Strengths:
- The critique of scaling oriented approach is timely and compellling.
- The authors support enchmark-driven scaling with three concrete examples.
- The recommendations are actionable and practical.

Weaknesses:
- Epistemical integration approaches are concept level, lacking concrete frameworks.

**Support:**

3

---

> ### Author Rebuttal · Authors · 2026-03-28
>
> We thank the reviewer for their appreciation of our approach and practical recommendations.
>
> Regarding epistemic integration, we intentionally avoid prescribing a one-size-fits-all framework, as language communities differ significantly in norms around data ownership, participation, and openness to ML integration. Rather than standardization, we advocate for adaptable, community-aligned practices.
>
> At the same time, our paper outlines several concrete and actionable mechanisms. In Section 4.2, we highlight successful industry-led efforts such as Meta’s Language Technology Partnership Program (LTTP), which has supported community-engaged work including Omnilingual ASR [1], as a model for responsible collaboration. In Section 4.3, we expand evaluation beyond standard metrics (e.g., BLEU, ROUGE) by pointing to community-centered initiatives such as UNESCO’s Language Technology for All (LT4All) [2], which provide structured platforms for native speakers and stakeholders to communicate and shape research directions. In Section 4.4, we emphasize the importance of sustained engagement through affinity spaces such as Indigenous in ML [3], as well as mentorship pipelines modeled after the ACL Student Research Workshop (SRW) [4], which would support emerging student researchers working closely with their language communities.
>
> Taken together, these mechanisms demonstrate that epistemic integration is already actionable through existing structures. Given the diversity of language communities, we believe flexible, community-driven approaches are more appropriate than imposing a single overarching framework.
>
> We believe this work is timely in surfacing critical challenges in endangered language revitalization, and in encouraging broader adoption of constructive, community-aligned practices. We hope this clarification addresses the reviewer’s concerns and supports a more favorable overall assessment. Thank you again for your thoughtful consideration.
>
> [1] Omnilingual ASR: Open-source multilingual speech recognition for 1600+ languages. arXiv, 2025.
>
> [2] International Decade of Indigenous Languages 2022–2032: Language Technologies for All (LT4All). UNESCO, 2025.
>
> [3] Indigenous in AI/ML: NeurIPS 2025 Affinity Event. NeurIPS, 2025.
>
> [4] Proceedings of the Association for Computational Linguistics 2025 Student Research Workshop. ACL, 2025.

---

> > ### Author Rebuttal · Reviewer_yKkf · 2026-04-01
> >
> > The authors have addressed my concerns, and I will maintain the positive score.

---

### Decision · Program_Chairs · 2026-04-30

**Decision:**

Accept (regular)

**Comment:**

All the referees are positive about the paper. I too agree that this is a very important position relative to endangered languages and their treatment by ML models. I vote that this paper be accepted as an oral presentation to enhance community awareness in addition to the technical discussion.